# The Potential of Composite Cements for Wound Healing in Rats

**DOI:** 10.3390/bioengineering11080837

**Published:** 2024-08-16

**Authors:** Alina Ioana Ardelean, Sorin Marian Marza, Madalina Florina Dragomir, Andrada Negoescu, Codruta Sarosi, Cristiana Stefania Novac, Cosmin Pestean, Marioara Moldovan, Liviu Oana

**Affiliations:** 1Department of Veterinary Surgery, Faculty of Veterinary Medicine, University of Agricultural Sciencies and Veterinary Medicine, 3-5 Manastur Street, 400372 Cluj-Napoca, Romania; alina-ioana.ardelean@usamvcluj.ro (A.I.A.); madalina.dragomir@usamvcluj.ro (M.F.D.); cosmin.pestean@usamvcluj.ro (C.P.); oanaliviu2008@yahoo.com (L.O.); 2Department of Veterinary Imagistics, Faculty of Veterinary Medicine, University of Agricultural Sciences and Veterinary Medicine, 3-5 Manastur Street, 400372 Cluj-Napoca, Romania; 3Department of Veterinary Pathology, Faculty of Veterinary Medicine, University of Agricultural Sciences and Veterinary Medicine, 3-5 Manastur Street, 400372 Cluj-Napoca, Romania; andrada.negoescu@usamvcluj.ro; 4Raluca Ripan Institute for Research in Chemistry, Babeș-Bolyai University, 30 Fantanele Street, 400294 Cluj-Napoca, Romania; liana.sarosi@ubbcluj.ro; 5Department of Microbiology, Immunology and Epidemiology, Faculty of Veterinary Medicine, University of Agricultural Sciences and Veterinary Medicine, 3-5 Manastur Street, 400372 Cluj-Napoca, Romania; cristiana.novac@usamvcluj.ro

**Keywords:** tissue engineering, composite cement, biomaterial, skin wound healing, biocompatibility

## Abstract

Recent developments in biomaterials have resulted in the creation of cement composites with potential wound treatment properties, even though they are currently mainly employed for bone regeneration. Their ability to improve skin restoration after surgery is worth noting. The main purpose of this research is to evaluate the ability of composite cement to promote wound healing in a rat experimental model. Full-thickness 5 mm skin defects were created, and the biomaterials were applied as wound dressings. The hybrid light-cured cement composites possess an organic matrix (Bis-GMA, TEGDMA, UDMA, and HEMA) and an inorganic phase (bioglasses, silica, and hydroxyapatite). The organic phase also contains γ-methacryloxypropyl-trimethoxysilane, which is produced by distributing bioactive silanized inorganic filler particles. The repair of the defect is assessed using a selection of macroscopic and microscopic protocols, including wound closure rate, histopathological analysis, cytotoxicity, and biocompatibility. Both composites exerted a favorable influence on cells, although the C1 product demonstrated a more extensive healing mechanism. Histological examination of the kidney and liver tissues revealed no evidence of toxicity. There were no notable negative outcomes in the treated groups, demonstrating the biocompatibility and efficacy of these bioproducts. By day 15, the skin of both groups had healed completely. This research introduces a pioneering strategy by utilizing composite cements, traditionally used in dentistry, in the context of skin wound healing.

## 1. Introduction

Biomaterials have been progressively evolving to regenerate or substitute body parts. The product must resemble the mechanical, physical, and chemical characteristics of the component it replaces, while being biocompatible, non-toxic, and sustainable [1,2]. Four generations of biomaterials are defined based on their features: first—bioinert and biostable, second—biocompatible and bioactive, third—biodegradable or bioresorbable, and fourth—biomimetic or bioinspired [3,4]. A biomaterial’s capacity to disintegrate gradually is crucial, ensuring that the organism easily assimilates it and restores any damaged cells [5,6,7].

Creating effective interactions to substitute, cure, or heal injured tissue is the purpose of biomaterials [3]. To successfully interact with a living system, a biomaterial must be biocompatible, safe to use, durable, and exhibit biomechanical and physicochemical consistency. Recently developed materials can influence the skin’s ability to heal and encourage cell division and the growth [2] of tissues, including dental, cutaneous, or skeletal cells.

In an earlier study, we evaluated in vitro two new composite cements: C1 and C2. Both products were described, and a cytotoxicity test was performed to evaluate their potential for healing. In vitro tests were conducted using palatal origin mesenchymal stem cells (MSCs), which exhibit prompt cell growth. As the reliability of the bioproducts was confirmed, further cutaneous in vivo tests were conducted, based on the cell viability exceeding 97% [8]. Even though the nature of the materials is suitable for tooth restoration, our objective was to investigate how our bioproducts behaved on the skin. While the foundation of both C1 and C2 is a mixture of monomers (Bis-GMA, UDMA, HEMA, and TEGDMA), their distinguishing feature is their filler content. Both biomaterials are comprised of hydroxyapatite and silica; however, C1 additionally contains quartz and barium glass (BaO), whereas C2 includes fluoroaluminosilicate glass and glass filler made of BaF2. Enhanced decomposition of the polycaprolactone core shell and excellent particle dispersion inside the polymeric matrix are ensured by the combination of mineral filler in the C1 specimen. In the meantime, C2 guarantees a more gradual insertion of the mineral filler particles into the polymer matrix. The bioproducts employed in the current research were selected based on their biological compatibility and capacity to promote cell attachment and growth.

Dental materials have undergone a tremendous evolution to ensure their success in clinical trials. Dental polymers and methacrylate monomers (such as Bis-GMA, HEMA, and UDMA) are utilized in various polymer resin-based dental products [9,10]. Strict compositional control allows for the precise balance of cytotoxic behavior and biocompatibility [11,12]. While there are limited reports of severe side effects, responses such as ophthalmic, cutaneous, and mucous membrane irritation; allergic dermatitis; stomatitis; and hepatic toxicity have been documented [13].

Hydroxyapatite (HAP) is a natural element occurring as calcium apatite [14], which is identified in skeletal tissues and teeth, and is employed as a ceramic in medicine, based on its biological activity [15]. It can be generated from coral [16], seashells [17], eggshells [18], fish or chicken bones [19], etc. Aside from its compatibility, HAP is additionally used to speed up soft tissue healing, and it possesses antibacterial characteristics.

Silica nanoparticles exhibit effectiveness in enhancing pharmaceutical delivery methods. They are frequently utilized as a filler product and can promote fibroblast growth, encourage the formation of collagen, and contribute to cell proliferation and transportation [20].

Barium is an alkaline metal present in tissues such as the enamel of the tooth. Used as a bioglass, it increases the production of angiogenic agents, including vascular endothelial growth factor (VEGF), which promotes vascular development, oxygenation, and nutrient delivery. Bioglass bioactivity stimulates cell adhesion and proliferation [21].

Vital biological resources can be used for developing novel methods and strategies in the field of skin wound healing. During rehabilitation, cutaneous tissue experiences a process of cell growth and reconstruction, as well as inflammation and angiogenesis. Research has demonstrated that the wound generates additional collagen throughout the healing process, along with growth factors, fibroblasts, and new blood vessels [22,23]. High-potential materials provide enhanced vascularity and angiogenesis [23,24]. The main goal of skin regenerative products is to prevent scars and encourage efficient healing throughout all stages. The four stages of wound healing are hemostasis, inflammation, proliferation, and remodeling [23]. The first phase starts on the third day, coinciding with the onset of angiogenesis and vascularization, as well as a high risk of infection [25]. Day three is when the likelihood of infection starts, posing a serious issue for the healing cycle. Each stage can be stopped or prolonged if the patient’s immune system is weakened, which is why we collected microbiological samples of the skin before surgery. The inflammatory reaction is prolonged if bacterial infection occurs [26]. It is feasible to achieve an antibacterial effect by employing filler particles. Our bioproducts lack bactericidal components and exhibit little antibacterial activity [8]. An extended inflammatory reaction can be caused if the bacterial overgrowth occurs beneath the bandage [26].

Animals remain one of the most accurate in vivo experimental models due to their ability to exhibit reflections of tissue healing mechanisms [22,23]; thus, Wistar–Lewis rats were used as the test subjects of interest.

The aim of the study was to evaluate the composite cements’ ability to promote wound healing on rat skin.

## 2. Materials and Methods

### 2.1. Animal Care and Use

In the present study, as a biological resource, we used 20 adult female rats, divided into two groups of 10 individuals per biomaterial (C1 and C2). The laboratory rat species, around 250 g in weight, belongs to the Wistar–Lewis line of the Muridae family [27]. This particular breed was selected due to its specific characteristics, such as an enhanced level of obedience, strong adaptability, and less susceptibility to microbial overgrowth [28,29].

The experiment was conducted at the Establishment for Breeding and Use of Laboratory Animals, Faculty of Veterinary Medicine, in Cluj-Napoca, Romania, after the animals were acquired from the Experimental Medicine Center of the University of Medicine and Pharmacy Iuliu Hatieganu.

In compliance with standards [29] the rats under study benefited from regular care and feeding conditions, which included a temperature of 23 °C, humidity of 55%, and light/dark cycles of 12 h each.

The experiment was approved by the Bioethics Committee of the University of Agricultural Sciences and Veterinary Medicine of Cluj-Napoca, no. 352/12.12.2022, and authorized by the Sanitary-Veterinary and Food Safety Department, Cluj-Napoca, through the project authorization no. 374/04.07.2023.

#### Surgical Procedure

The rats were anesthetized, based on their respective weights, with a combination of ketamine (50 mg/kg Narkamon Bio, Bioveta, Ivanovice na Hané, Czech Republic) and medetomidine (0.25 mg/kg Domitor, Biotur, Teleorman, Romania), injected intraperitoneally.

Postoperative, buprenorphine was used subcutaneously for analgesia (1 mg/kg Bupaq, Biotur, Teleorman, Romania) on the first day, followed by meloxicam (2 mg/kg meloxidolor 5%, Vetro, Romania) for three days, one dose per 24 h.

Following anesthesia, blood was collected from the infraorbital sinus of each rat to assess hematological and biochemistry factors.

The next step was to prepare the skin for the procedure. First, the elected area was established and the hair was trimmed on the dorsal part of the thorax from T1 to L1 [30,31]. To guarantee total hair removal, a specific cream was applied and left on for only three minutes to avoid burning the skin [32]. After the cream was removed, the antisepsis was performed using 4% chlorhexidine and 70% sanitary alcohol.

A total of 20 rats were used. They were divided into two batches. Two dermal excisions were created for each rat on both sides of the spine. The highly developed panniculus carnosus muscle was prevented from contracting using a 1 mm semi-hard silicone thread that was attached to the skin [30]. The internal hole had a diameter of 6 mm, and the external hole had a diameter of 15 mm. The rings were sterilized before use.

To execute the cutaneous defects, several steps were followed, as described: the rodent was placed in lateral recumbency. Afterwards, using two hemostatic clamps, a cutaneous flap was created, dorsal to the spine. Using a 5 mm biopsy punch that was positioned around 8 mm from the skin folds, the defects were acquired by cautiously twisting the biopsy punch to penetrate through both layers of the skin. Therefore, two identical excisions were made, 16 mm apart. Sterile wooden stick cotton swabs were used to clean the surgical area. The next step was to secure the silicone ring to the skin using an adhesive gel and four sutures, placed cardinally.

The biomaterials were then applied to the dermal defect as the protocol’s next step. The left defect served as the control, and no product was added, while the experimental biomaterial [8] was applied to the right defect. To facilitate the application, the composite cements C1 and C2 were synthesized as a paste for this study.

Sterile bandages, secured with PEHA-fix (Hartmann, Ilfov, Romania), were placed over the defects to prohibit auto-mutilation and prevent infection of the wounds.

Photographs were captured on the 1st, 3rd, 6th, 11th, and 15th day, and the product was reapplied on the 6th day. The wounds were covered with bandages until the very last day of the trial. Since the procedure for dressing and photographing the defects was not one that caused pain, inhalation anesthesia with Isoflurane (Anesteran 99.9%, Rompharm, Otopeni, Romania) was chosen.

After the surgical region was harvested, samples from the skin, kidneys, and liver were collected on the 7th and 15th day. Additionally, blood samples were also obtained.

The research concluded when the skin wound had fully healed on the 15th day. An overdose of anesthesia and cervical dislocation were used to euthanize the animals

### 2.2. Composite Cement Sample Preparation

The samples used in the study have an organic matrix and an inorganic phase. The experimental cementing materials C1 and C2 were obtained by dispersing bioactive silanized inorganic filler particles in the organic phase. The inorganic phase was treated with γ-methacryloxypropyl-trimethoxysilane (A174) (Sigma Aldrich, Darmstadt, Germany) to enhance compatibility with the organic matrix, based on mixtures with different particle sizes. The samples were also supplemented with camphorquinone (0.5% relative to the liquid mixture)/amine (1%) as the initiator/activator. The composition of both products is illustrated in Table 1. It is important to highlight that we used all the materials listed in the table, but we deliberately skipped the polymerization procedure.

The two biomaterials (C1 and C2) were developed at the Raluca Ripan Institute for Research in Chemistry. The type of mineral filler employed determines the specific properties of the bioproduct. The matrix/filler ratios are 65/25% for both C1 and C2. The fillers were encapsulated within polycaprolactone (PCL) microcapsules prepared in our lab, using buffalo whey as a dispersant.

Regarding the composition and features of the biomaterials, the following elements comprise the matrix of C1: Bisphenol A (glycerolate dimethacrylate), UDMA (Urethane dimethacrylate), HEMA (2-Hydroxyethyl methacrylate), and triethyleneglycol-dimethacrylate. The components of the filler include silica, nanostructured hydroxyapatite, silanized quartz, and BaO glass. The microstructural specifications are as follows: the silanized quartz and nanostructured hydroxyapatite clusters are of submicron size, while the barium oxide glass (BaO) particles range in diameter from 1 to 10 μm. Carbon (C) and oxygen (O) represent the majority of the elemental composition of the organic matrix, with calcium (Ca), silica (Si), phosphorus (P), and barium (Ba) present in amounts that are similar to those in the nanostructured hydroxyapatite.

Biomaterial C2’s matrix composition is the same as C1’s. The filler component includes BaF2 glass, nanostructured hydroxyapatite, silica, and fluoroaluminosilicate glass. The microstructural characteristics are as follows: fluoroaluminosilicate glass with a diameter of 0.04–0.50 μm, barium glass (BaF2) particles ranging in diameter from 2 to 6 nm, and nanostructured hydroxyapatite clusters of submicron size. The elemental composition is similar, but Silica (Si), barium (Ba), phosphorus (P), and calcium (Ca) are present in lower levels than those found in C1, suggesting a reduced percentage of nanostructured hydroxyapatite. The polymer matrix accounts for the majority of the composition, which is comprised of carbon (C) and oxygen (O).

We provided a broad overview of the bioproducts in our earlier remarks, but Ardelean et al.’s study (2023) provides an in-depth analysis and documentation of the microstructural features and an elemental description of the cements [8].

### 2.3. Bacteriological Assays

After the swabs were obtained on the first and seventh day of the study, the bacterial analysis was carried out at the Microbiology, Immunology, and Epidemiology Laboratory of the Faculty of Veterinary Medicine Cluj-Napoca, Romania. On day one, the sampling for the bacteriological examination was carried out after trimming the hair in the area where the defects were to be made, prior to performing antisepsis. On the seventh day, the swabs were collected from the wound margins.

Sterile swab samples were obtained and processed on the same day. They were inoculated on Mueller–Hinton agar and Columbia sheep blood agar and incubated for 24 h at 37 °C under aerobic conditions. Following incubation, the samples were analyzed, and typical Gram staining was used to assess the bacterial morphology (shape, arrangement) and tinctorial features. The cultural characteristics of bacteria grown on culture media, including colony size, shape, color, consistency, and the presence or absence of hemolysis on blood agar, were assessed. Moreover, to preliminarily identify Gram-positive cocci, the slide catalase test was performed using 3% hydrogen peroxide.

Further analysis was carried out on the obtained isolates to identify them at the species level. Therefore, bacterial identification was performed using the Vitek^®^ 2 Compact system (Biomérieux, Marcy-l’Etoile, France). This computerized approach analyzes 64 biochemical characters and compares the results to a database to phenotypically identify the bacterial species.

### 2.4. Blood Tests

Hematology and biochemistry components were additionally investigated to prove the immunological state of the rats on the first and seventh days.

After the rats are anesthetized, the skin surrounding the eyes is pulled tight, and the animal is rubbed with the thumb and fingers of the nondominant hand. A capillary tube is placed at a 30-degree angle to the nose in the medial canthus of the eye, and a small amount of thumb pressure is enough to collect blood [33]. Specific blood collection tubes are required. After drawing the necessary amount of blood from the plexus, the capillary tube is carefully removed, and gentle pressure is applied locally to stop the bleeding [34]. The findings were then compared with those from the specialized research literature [35,36].

#### 2.4.1. Hematology Blood Tests

The blood was processed using an Abaxis VetScan HM5 hematology analyzer (Abaxis Inc., Union City, CA, USA) to calculate the complete blood count [37] on whole blood: WBC: white blood cells; LYM: lymphocytes; MON: monocytes; NEU: neutrophils; RBC: red blood cells; HCT: hematocrit; HGB: hemoglobin; MCV: mean cell volume; MCH: mean corpuscular hemoglobin; MCHC: mean corpuscular hemoglobin concentration; PLT: platelet count.

#### 2.4.2. Biochemistry Blood Tests

The blood was processed using an Automatic Veterinary Chemistry Analyzer Element RC (Scil Animal Care Company, Alfort, France) to determine the following parameters [38]:

ALB: albumin; TP: total protein; TB: total bilirubin; ALT: alanine aminotransferase; ALP: alkaline phosphatase; CREA: creatinine; UREA: blood urea nitrogen; GLU: glucose; CA: calcium; PHOS: phosphorus; K: potassium; NA: sodium.

### 2.5. Histopathological Analysis of Skin Defects

This study was conducted on 20 rodents, divided into two batches of 10 individuals per biomaterial (C1 and C2). Considering two different time frames, five individuals on day 7 and five on day 15 were sacrificed to harvest the skin, liver, and kidneys for histopathological examination. The samples were then fixed in 10% buffered neutral formaldehyde for 24 h. Following fixation, the samples underwent standard histopathological processing. The paraffin-embedded tissues were sliced into 2-micrometer-thick sections and stained with hematoxylin and eosin (H&E) and Masson’s trichrome (MT) stains for evaluation. The slides were examined by two different pathologists using an Olympus BX41 (Olympus Europa SE&Co, Hamburg, Germany) microscope. The images were captured with an Olympus UC 30 digital camera (Olympus Europa SE&Co., Hamburg, Germany) and processed using a special image acquisition and processing tool: Olympus Stream Basic.

On days 7 and 15 of the healing phase, the sample was examined and compared. Both the C1 and C2 groups were compared with the Blank group. Additionally, liver and kidney tissues were collected to determine if the use of the biomaterial was inducing any acute toxicity.

### 2.6. Measurements of Wound Size Reduction

To evaluate the dynamics of wound healing, photographs were taken on the 1st, 3rd, 6th, 11th, and 15th day. The images were captured with an iPhone 14 Pro, Macro Mode, 48 MP. To assess cutaneous repair, we utilized the ImageJ 1.53 [39] free software with the scale set to 10 units per millimeter. Transforming the images to grayscale, we created a map displaying a three-dimensional histogram of the pixel distribution around the injury. Utilizing the primary defect area (A0) and the wound defect area at each time point (At), the degree of the healing was determined by applying the following formula:(1)$$\mathrm{Wound}\;\mathrm{closure}\;(\%)=\frac{A_{0}-A_{t}}{A_{0}}\times 100.$$

### 2.7. Statistical Analysis of the Wound Closure

The mean and standard deviation were used to report all tissue regenerating data. The statistical values were acquired using the GraphPad Prism 8.0 program. For wound healing, the Student’s *t*-test, along with one- and two-way analyses of variance (ANOVA), were conducted [40,41].

All data reported a statistical significance when comparing the Blank with the C1 group. Translated to clinical importance, our biomaterial C1 is proven to encourage the healing process, with a significance of *p* < 0.0001, and on the 15th day, the value is *p* < 0.01. Between the Blank and C2 groups, there was no statistical significance, yet on the 3rd day, the value is *p* < 0.05.

### 2.8. Statistical Analysis of the Hematological and Biochemical Analytes

The mean ± SD (standard deviation) of the measured parameters was determined independently for days 1 and 7 of the trial using GraphPad Prism 8.0 [42]. The blood parameter values were independently evaluated for statistical significance using one- and two-way ANOVA. *p* < 0.05 was deemed to be statistically significant. The minimum and maximum values for each parameter were also provided as a reference [43,44].

## 3. Results

### 3.1. Bacteriological Assays

For the purpose of identifying any opportunistic pathogens that could have an effect on the healing process, a qualitative assessment of the microbial flora (bacterial species identification) was conducted during the research.

Preliminary analysis of the plates revealed that all processed samples exhibited a homogenous and similar bacterial flora, with a significant prevalence of germs from the *Staphylococcus (S.) intermedius* genus. Macroscopic evaluation revealed circular, smooth, medium-sized, white-colored (Figure 1A) colonies on the blood agar (Figure 1B). Smears were acquired from the culture plate, stained using the Gram-staining technique, and inspected microscopically. Microscopic research revealed the presence of *Staphylococcus intermedius*, a saprophytic skin germ, but no negative consequences were observed for the rats.

### 3.2. Blood Tests

Blood tests were performed to determine whether the rats exhibited any health issues and if they could be included in our study. To thoroughly assess the biological suitability and systemic impact of the bioproducts, we included data obtained through biochemical and hematological analyses. The data confirmed the biocompatibility and systemic safety of the biomaterials.

#### 3.2.1. Hematological Profile

The complete blood counts of all groups indicated normal levels, which were comparable to those in the control group, and there were no statistical variations. Table 2 shows the merged results of the hematological profiles from days 1 and 7 of the trial. The utilization of bioproducts had no harmful effect on blood hematology.

#### 3.2.2. Biochemical Profile

For the biochemical profile, whole blood was tested to determine the parameters subsequently detailed in Table 3. The results of biochemical tests revealed that there were no liver or renal dysfunctions or any other disturbances.

### 3.3. Skin Healing Assessment

When the epidermal barrier is breached, the cellular and chemical responses inside the skin layers will synchronize to begin the recovery process. Access to the skin creates ideal circumstances for germs to colonize and reproduce uncontrollably. Skin healing was assessed through macroscopic screening. Optical pictures of the wound were captured from the first day to the fifteenth day (Figure 2). Subtle compositional variations between the C1 and C2 groups are also evident in regards to the coloration of the resulting bioproducts.

The macroscopic examination of the skin’s rehabilitation illustrates that the C1 biomaterial regenerates the skin more rapidly than the C2 bioproduct. The healing process revealed that on the third day after surgery, the sealing of the cutaneous defect began, and crusts were visible in all groups. On the sixth day, the skin closure became increasingly noticeable and more obvious in the C1 group. Even though they were all still visible on day 11, the defects had diminished, leaving a thin crust above the wound. Fifteen days following the surgery, the healing phase was finished, and a scar replaced the original location of the defect. At that point, there was no visible distinction among the scars. The bioactive elements of the cement stimulated re-epithelialization and tissue remodeling.

For every image that was examined, a surface graph was created to provide a visual representation of the healing (Figure 3).

Overall, the C1 group’s results were better than those for the C2 group. It was evident that there had been improvements from the third day forward. Both bioproducts stimulate wound repair in rats, without leading to tissue necrosis. It should be noted that, by the end of the experiment, the skin defect had healed in all groups.

### 3.4. Percentage of Wound Closure

The percentage of wound closure was determined and applied to compare the outcomes between the Blank, C1, and C2 groups. In comparison to those treated with C2, the content regenerates more quickly in the first week of therapy for those treated with C1 (Figure 4). This suggests that the C1’s mineral filler content impacts the rate at which the healing process develops. Therefore, future research featuring immunocompromised subjects who could benefit from this prospective usage should be conducted.

### 3.5. Histopathological Analysis of Skin Defects

On day 7, the histological examination of rat tissue sections treated with C1 and C2 revealed several morphological changes. Those included the formation of granulation tissue with pronounced vascularization (Figure 5), characterized by the formation of new blood vessels lined by plump endothelial cells (neoangiogenesis) and plump mesenchymal cells, confirmed by MT staining. Additionally, a mild inflammatory infiltrate comprised of neutrophils, macrophages, and occasional lymphocytes, alongside mild hemorrhage, were observed at the site of the defect. Within and around the granulation tissue, multifocal, amorphous, pale eosinophilic material, with slight crystallization, was present. The material consisted of powdered substances intermixed with macrophages, multinucleated macrophages, and neutrophils. The surface epidermis was present at the peripheral regions of the defect but absent from its central area, potentially attributed to the application of additional material on Day 6.

On day 15, the control group used for the C1 biomaterial presented the following morphological aspects: the defect area was replaced by partially organized granulation tissue, consisting of abundant mesenchymal cells, collagen fibers, and blood vessels (Figure 6A), confirmed by MT staining (Figure 7A). Within the granulation tissue, multifocal discrete lymphoplasmacytic inflammatory infiltrate and rare siderophages, mainly in the deeper regions of the scar tissue, were noted (Figure 6C).

On day 15, the tissue sections from the rats treated with C1 material showed complete defect healing. The dermis was replaced by fibrous tissue (Figure 8), composed of fibroblasts aligned parallel to the epidermis, vascular channels with erythrocytes present in the lumen (Figure 6B), and a scattered inflammatory infiltrate primarily composed of lymphocytes and plasma cells, features confirmed by MT staining (Figure 7B). Additionally, partial restoration of sebaceous units was identified at the defect site. Focally, in the deep dermis, within the cytoplasm of some macrophages, fragmented biomaterial was observed, without any biomaterial in the extracellular space, proving its complete resorption. Rare multinucleated macrophages were noted at this level (Figure 6D).

In the control group of the second biomaterial, the main morphological changes, observed on day 15 at the defect level, were characterized by complete healing of the epidermis, with all the layers visible (Figure 9A), with the proliferation of fibroblast, collagen fibers, blood vessels (Figure 7C) and mild lymphoplasmacytic inflammatory infiltrate in the dermis (Figure 9C). No regeneration of sebaceous units was identified in the evaluated samples.

In the group treated with C2 material, complete wound healing was noted on day 15. The dermis was composed of a moderate number of fibroblasts oriented parallel to the epidermis, intertwined with blood vessels and collagen fibers like the ones from the normal tissue (Figure 9B), along with a moderate lymphoplasmacytic inflammatory infiltrate, features confirmed by MT staining (Figure 7D). Within the deep dermis, moderate to severe pyogranulomatous inflammation was observed. Within the inflammation, a high number of macrophages, as well as multinucleated macrophages, were identified, centered on a large island of amorphous eosinophilic material, denoting an incomplete resorption of the biomaterial. Furthermore, the material was observed in the cytoplasm of the cells mentioned previously (Figure 9D). Moderate infiltration with lymphocytes and plasma cells was identified at the periphery of the granulomatous reaction. The wound was covered by a healed epidermis, including all skin layers. No new sebaceous units were identified in the samples evaluated.

In conclusion, complete healing of the wound was observed in both groups, although in the group treated with C2, a more severe foreign body reaction was identified, with incomplete resorption of the biomaterial, compared to that noted in the group treated with C1, in which complete resorption of the biomaterial and the healing of the sebaceous units were noted.

The kidney and liver samples underwent histopathological examination. It is essential to determine suitable precautions to avoid organ damage as soon as cellular alterations caused by such events are detected.

Histological examination revealed that the hepatocyte microstructure remained intact. No abnormalities were found in the hepatic tissue of medicated rats, including fibrosis, destruction of hepatocytes, necrosis, or lymphocyte agglomerates caused by inflammatory responses. Additionally, no abnormal changes in renal tissue, such as fibrosis, reduction in duct cells, disappearance of glomeruli, necrosis, or lymphocyte clusters caused by inflammatory responses, were observed. In conclusion, no significant findings were noted in the liver or the kidneys for either C1 or C2 on day 15 (Figure 10).

## 4. Discussion

Although the prediction of undesirable medication effects has improved in recent years, many novel pharmaceuticals, with unique pharmacological pathways, continue to pose important challenges. This is especially true for biotherapeutics and their drug-induced immunological responses, spontaneous organ toxicity, or systemic toxicity. Furthermore, healthy animals may serve as a model, in certain circumstances, for predicting susceptibility to adverse consequences [45]. Preclinical studies commonly evaluate general toxicity and targeted organ toxicity, as well as product absorption, dissemination, metabolization, and elimination [46,47]. Several paraclinical investigations were conducted at the beginning of this study. The goal was to determine whether the rodents were in satisfactory condition for generating realistic findings and to discover if our bioproducts would interfere with their clinical status, locally or generally. Specifically, blood tests were conducted as part of the initial study.

Moderate and elevated doses of anesthesia or analgesia can alter several blood biochemistry and hematological parameters; these dosages can also impact body mass index and dietary habits [48]. While local or systemic conditions impact the healing process of cutaneous tissue [49], during anesthesia, elevations in hematological or biochemical parameters that can interfere with the dynamics of the research can also be noted [50]. Medetomidine can reduce white blood cell (WBC) counts, while increasing HCT and HGB levels. Other research suggests that administering ketamine can increase AST and ALT levels [51,52]**,** and stress hyperglycemia was also noticed [53]. Our subjects behaved well under anesthesia, without any complications, and blood tests revealed normal ranges. 

Typically, drug use exerts unfavorable consequences on the body’s defenses. Even minimal alterations, such as a skin wound, might result in catastrophic changes in an immunosuppressed body. Healthy skin attempts to maintain its integrity by regulating the skin contents from becoming hazardous to the skin or by limiting any infiltration of the tissues below. A delay in lesion healing could potentially be the result of a local trigger (infection), a systemic trigger, or both [49]. Bacteriological swabs were collected to assess the microorganism population. Microbial overgrowth is a notorious phenomenon that interferes with skin healing [49]. Studies indicate that the majority of acute lesions exhibit more Gram-positive than Gram-negative microbes [54]. Cumulatively, general data demonstrate that a non-physiological incident might trigger the proliferation of the commensal and saprophytic skin microbiota and disrupt the normal healing process. The isolated germs had no impact on wound healing in this trial. Combining blood tests and bacteriological discoveries, we established that the body was capable of overcoming our external intervention and healing adequately.

The chemicals in the biomaterials can disrupt the delicate balance of the body’s immune system. For instance, consequences can include alterations to the synthesis of cytokines, which might affect the beginning and management of inflammation [55]. Throughout the current investigation, the skin reactivity was tested throughout the application of the composite cements. Earlier in vitro studies indicated significant cell viability and low cytotoxicity when the cements were employed on mesenchymal stem cells, giving us adequate evidence to continue their investigation. Even though the biomaterial composition is more suitable for skeletal tissue, the bone is in intimate contact with muscle and connective tissue, and during surgeries, the cutis is also impacted. As explained in a previous section, the biocomposite include two major components: a matrix (monomers and polymers) and a mineral filler. Residual monomers and polymers have been detected in dental restorations and are often released into the mouth, generating an extensive variety of negative health consequences, both locally or systemically [56]; these residuals have been recognized as major allergens [9]. Depending on the concentrations and quantity, these residuals can substantially activate cytotoxicity-triggering mechanisms, leading to apoptosis and necrosis [10]. The mineral element includes nanohydroxyapatite, the most significant constituent of human bone [57,58]. The regular distribution of hydroxyapatite units inside the C1 and C2 biomaterials results in considerable cell growth, as demonstrated in the research of Ardelean et al. [8]. The literature suggests that the bio-integration of implants leads to an increase in alkaline phosphatase levels [59,60]. Both BaO and BaF_2_ have been employed, given their ability to establish powerful chemical interactions with soft and hard tissue, providing excellent angiogenic qualities, which were demonstrated in our study as well. C2 was proven to be more effective than C1 in the in vitro study, while C1 contains more bioactive minerals, leading to increased cell viability and alkaline phosphatase reaction. Our research determined that the C1 composite cement significantly improved skin healing. The mineral filler and volume are crucial for bioproducts to achieve successful regenerative outcomes. It is expected that tiny variations in the proportion of the material composition could significantly impact the product’s biological behavior, according to the information available in the literature [61]. A surface graph was created for each evaluated picture in order to visually show the resulting healing [36,37]. We did not employ the last step of polymerization in the creation of our biomaterials. This is another valuable study finding. Unpolymerized bioproducts are known to cause skin sensitivity or irritation due to their chemical structure. More importantly, the materials’ mechanical strength and longevity were altered. To prove that polymerization is a property that greatly affects the material’s deterioration, we are carrying out more research. The purpose of these investigations is to clarify how polymerization affects the material’s durability and structural stability. Our objective in this study was to evaluate whether biocompatibility and potential negative effects like necrosis or apoptosis were present. In order to successfully conclude this research and obtain a thorough comprehension of the material’s performance, further investigations are required. Nonetheless, we see this as a fundamental place to start.

The multiple adverse effects, whether local or general, demonstrated in the early research led us to investigate not just the local skin response, but also organ-induced toxicity [62]. The liver and kidneys were evaluated. A qualitative histological investigation of these tissues and an analysis of the parenchymal cells were conducted. These examinations indicated that no abnormal alterations occurred following the research interval. The structural integrity of the hepatocytes and blood sinusoids was maintained in the livers of the treated rats, and no pathological alterations were observed in the renal tissue. Furthermore, biocompatibility was examined during the acute exposure investigation employing a restricted number of Wistar rats, which can be considered a study limitation.

The study’s small number of individuals was chosen to minimize oversampling, and a comprehensive investigation of tissue histology strengthened the conclusions. Another limitation we might consider is the defect’s modest size. In a future investigation, we may try to test our bioproducts on large skin abnormalities which may be associated with many illnesses. Furthermore, our rodents were considered healthy subjects. We may consider evaluating the products on sick individuals in future trials. This study also has limitations in terms of the lack of molecular analysis, ultrastructural examinations, or immunocytochemical studies.

Despite exposure to dental materials, tissue morphology in the skin, liver, and kidneys was not significantly altered, indicating that the dental cement employed is biocompatible. Nonetheless, this work offers a new perspective on the effects of composite cements on skin healing. Although the tissue architecture of rats is relatively equivalent to that of humans, it is challenging to extrapolate how the findings will be applied in people.

## 5. Conclusions

The bacteriological examination of the skin indicated that there were no pathogenic microorganisms that could interfere with skin healing.

Additionally, blood testing showed that the rodents were healthy, and their status did not impede the skin’s ability to heal.

According to the macroscopic evaluations, biomaterial C1 promoted wound closure on the third day and presented superior qualities when compared to those of biomaterial C2, for which wound closure began on day six.

Furthermore, the histological analysis validates the better performance of bioproduct C1, for which the sebaceous units were partially restored, neoangiogenesis was evident, and the composite cement C1 was completely absorbed, compared to the results of the C2 material, for which no neoangiogenesis was observed, and the inflammatory reaction was still present.

Histological investigations of the organs demonstrated the absence of hepatotoxicity and nephrotoxicity after 15 days of topical application of the materials, validating the biocompatibility and reliability of both composite cements.

Despite modest compositional differences, C1 and C2 biopolymers demonstrated distinct skin healing benefits. These encouraging results establish a model for further studies.

In essence, our work highlights the biocompatibility of two new nanostructured dental materials, making it a promising choice for further research.

## Figures and Tables

**Figure 1 bioengineering-11-00837-f001:**
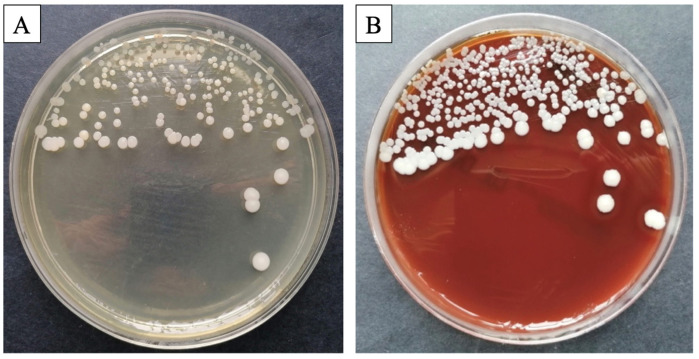
Representative images of bacterial colonies (**A**), processed on blood agar (**B**).

**Figure 2 bioengineering-11-00837-f002:**
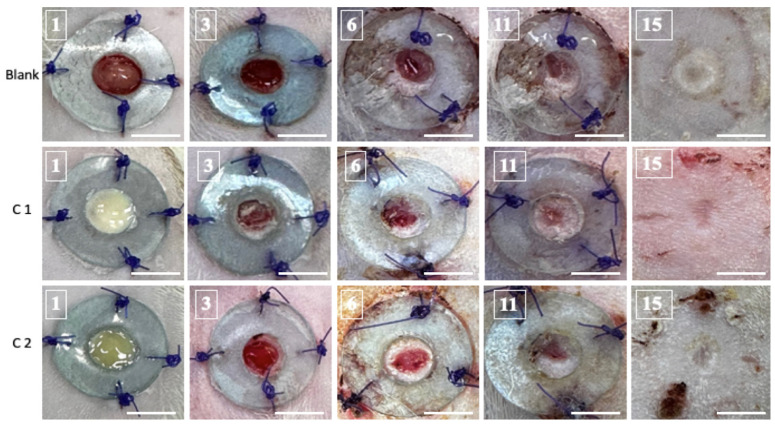
Representative images of full-thickness skin defects in Blank, C1, and C2 rats at 1, 3, 6, 11, and 15 days post-surgery (scale bar: 6 mm).

**Figure 3 bioengineering-11-00837-f003:**
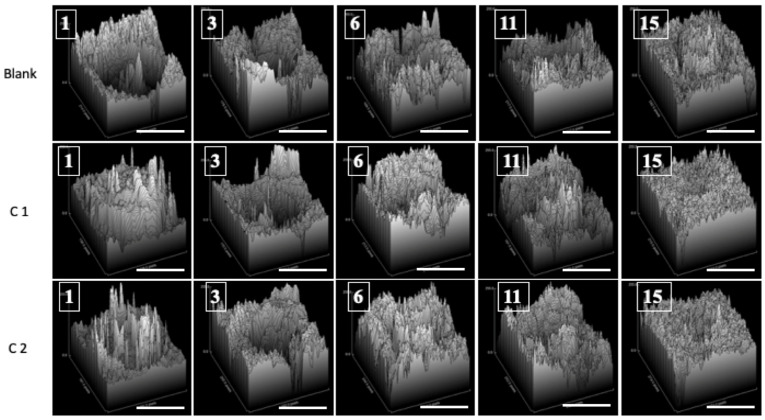
The evolution of wound healing is presented in 3D at 1, 3, 6, 11, and 15 days post-surgery (scale bar 6 mm).

**Figure 4 bioengineering-11-00837-f004:**
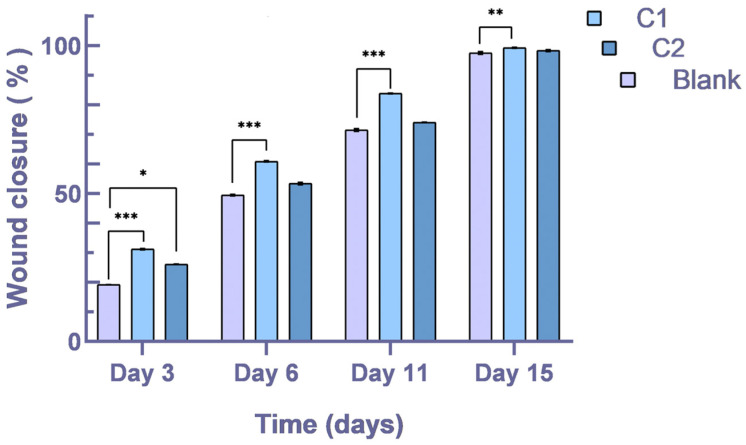
Percentage of wound closure of the defect treated with C1 and C2 composite cements. Statistical evaluations performed using one- and two-way ANOVA; * *p* < 0.05; ** *p* < 0.01; *** *p* < 0.001.

**Figure 5 bioengineering-11-00837-f005:**
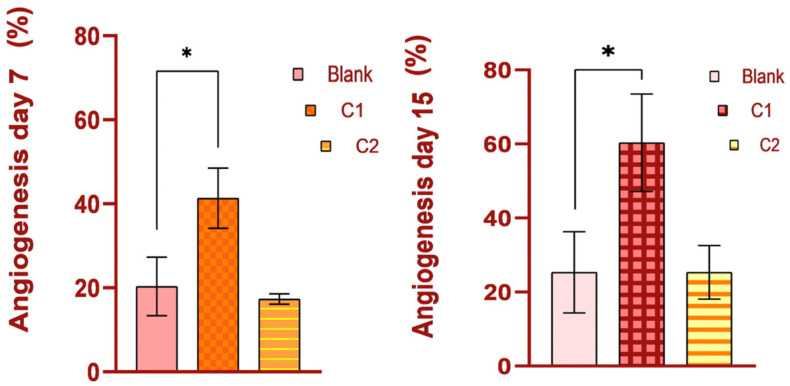
Angiogenesis (%) on day 7 versus day 15.; pronounced vascularization in C1 groups; * *p* < 0.05.

**Figure 6 bioengineering-11-00837-f006:**
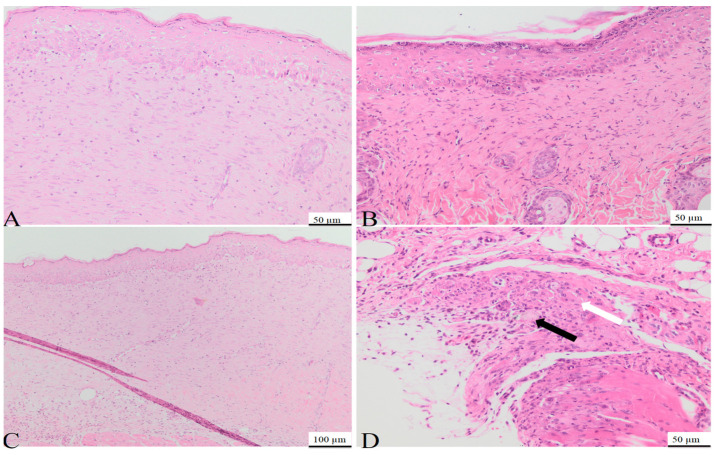
Photomicrographs of skin and subcutis obtained on day 15 from the control group (**A**,**C**) and from rats treated with the C1 biomaterial (**B**,**D**). In the control group, the defect is filled with fibrous tissue composed of fibroblasts oriented parallel to the epidermis, collagen fibers, and blood vessels, and covered by a completely regenerated epidermis (**A**), H&E, 50 µm; multifocally, predominantly in the deep dermis, scattered lymphocytes and plasma cells are present within the scar tissue (**C**), H&E, 100 µm; (**B**): the defect is replaced by fibrous tissue composed of fibroblasts oriented parallel to the epidermis, mature collagen fibers, scattered inflammatory cells, mainly lymphocytes and plasma cells, and regenerated sebaceous units covered by a completely regenerated epidermis (**B**), H&E, 50 µm; (**D**): fragmented biomaterial present within the cytoplasm of the macrophages (black arrow) and rare multinucleated macrophages (white arrow) within the deep dermis, (**D**): H&E, 50 µm.

**Figure 7 bioengineering-11-00837-f007:**
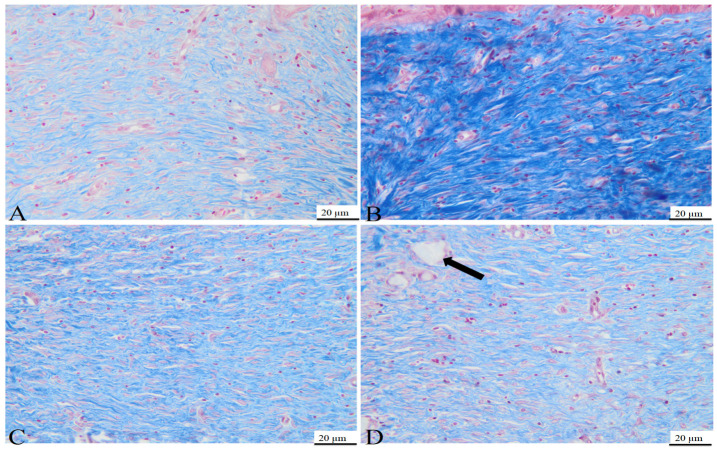
Photomicrographs of the dermis obtained for the area of the defect at 15 days post-surgery in C1 control case (**A**), C2 control case (**C**), rats treated with C1 biomaterial (**B**), and rats treated with C2 biomaterial. Biomaterial fragments (black arrow) (**D**); MT, 20 µm.

**Figure 8 bioengineering-11-00837-f008:**
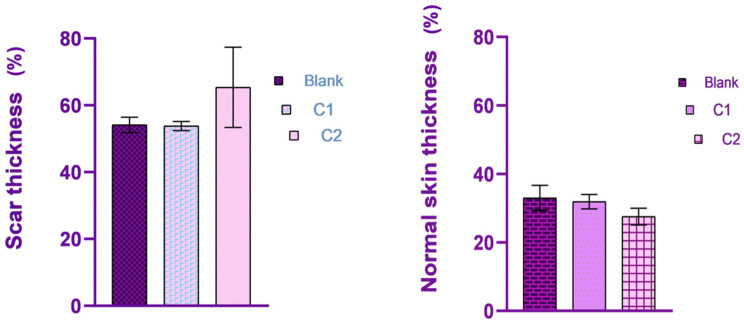
Normal skin thickness (%) compared with scar thickness (%) on day 15.

**Figure 9 bioengineering-11-00837-f009:**
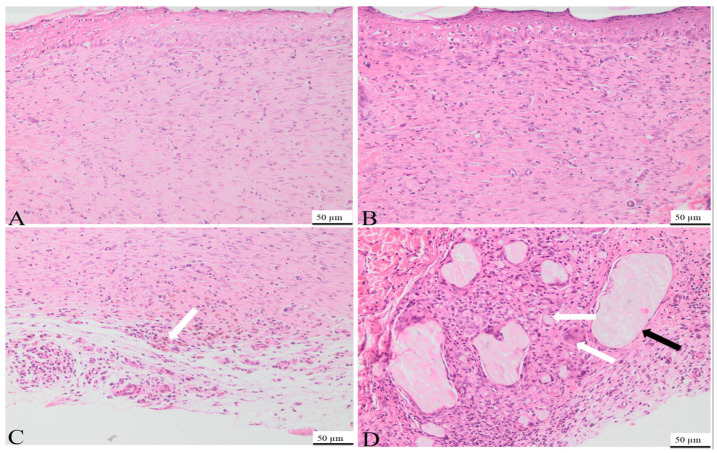
Photomicrographs of the skin and subcutis obtained on day 15 from the control group (**A**,**C**) and from rats treated with C2 biomaterial (**B**,**D**). The defect is replaced by mesenchymal cells oriented parallel to the regenerated epidermis, blood channels, and collagen fibers (**A**); H&E, 50 µm; within the deep dermis, a mild inflammatory infiltrate, mostly lymphocytes, plasma cells, and rare siderophages (white arrow) are present (**D**), H&E, 50 µm. (**B**): in the group treated with C2 biomaterial, the defect is filled by fibrous connective tissue, composed of spindle cells, blood vessels, collagen fibers, and scattered inflammatory cells (**B**); the pyogranulomatous reaction can be noted multifocally within the dermis, represented by macrophages, multinucleated macrophages (white arrows), and neutrophils centered on biomaterial fragments (black arrow), H&E, 50 µm.

**Figure 10 bioengineering-11-00837-f010:**
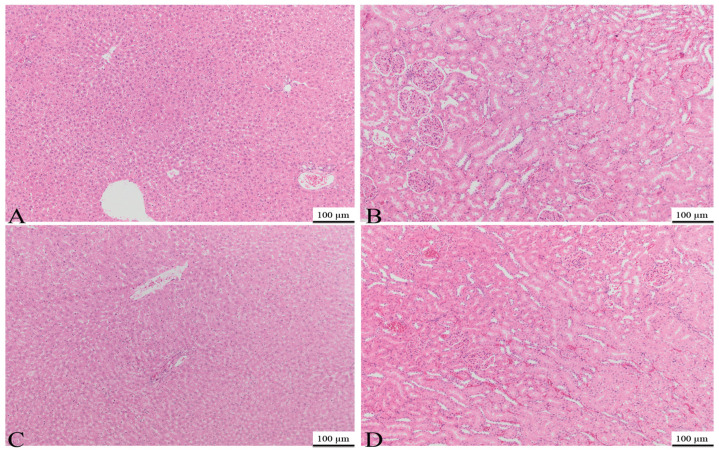
Photomicrographs of the kidney and liver obtained on day 15 from the C1 group (**A**,**B**) and C2 group (**C**,**D**), with no significant findings noted; H&E, 100 µm.

**Table 1 bioengineering-11-00837-t001:** Experimental cement composition.

Biomaterials	Manufacturer	Monomers	Filler Content
hybrid composite cement (C1)	UBB-ICCRR, Cluj-Napoca Romania	25%, Bis-GMA; UDMA; HEMA TEGDMA	65 wt%, HA (particle size 0.01–0.06 μm and 5–8 nm); silica, barium glass (BaO) (particle size 0.01–0.035 μm and 2–6 nm); quartz
hybrid composite cement (C2)	UBB-ICCRR, Cluj-Napoca Romania	25%, Bis-GMA; UDMA; TEGDMA	65 wt%, HA (particle size 0.01–60 μm and 5–8 nm); silica, glass filler (with BaF2) (particle size 2–6 nm); fluoroaluminosilicate glass (0.04–0.50 μm)

Bis-GMA-2,2-bis(3-(2′-hydroxy-3′methacryloyl-oxypropoxy)phenyl) propane; HA-hydroxyapatite; glasses (synthesized in UBB-ICCRR laboratory, Cluj-Napoca, Romania); TEGDMA-triethyleneglycol-dimethacrylate (Aldrich, Steinheim, Germany); DMAEM-2-dimethyl(aminoethyl)methacrylate (Aldrich, Steinheim, Germany); Cq-camphorquinone (Aldrich, Steinheim, Germany).

**Table 2 bioengineering-11-00837-t002:** Combined mean values and standard deviations (SDs) presented for all hematological parameters tested on days 1 and 7, as there were no notable variations. Statistical evelautions were performed utilizing one- and two-way ANOVA.

Analyte	Blank Mean	Blank SD	C1 Mean	C1 SD	C2 Mean	C2 SD	References
WBC 10^9^ cells/L	7.25	0.05	8.13	0.13	7.18	0.09	2.10–19.50
LYM 10^9^ cells/L	4.83	0.05	5.46	0.13	5.36	0.16	2.00–14.10
MON 10^9^ cells/L	0.19	0.03	0.25	0.15	0.33	0.02	0.00–0.98
NEU 10^9^ cells/L	2.24	0.05	2.60	0.14	1.89	0.06	0.10–5.40
LYM %	66.93	0.37	68.19	1.24	76.79	1.26	0–100
MON %	2.84	0.45	3.12	1.74	4.93	0.34	0–100
NEU %	30.27	0.78	30.19	1.15	22.43	1.29	0–100
RBC 10^12^ cells/L	8.46	0.02	7.98	0.12	7.98	0.11	5.30–10
HGB g/dL	16.34	0.06	15.93	0.27	16.12	0.31	14–18
HTC %	43.36	0.21	42.77	1.26	43.66	1.04	35–52
MCV fL	51.84	0.00	54.17	1.03	56.34	0.42	50–62
MCH pg	19.55	0.17	19.85	0.29	20.86	0.15	16–23
MCHC g/dL	37.62	0.13	37.09	0.28	38.66	0.12	31–40
PLT 10^9^ cells/L	641.25	5.91	665.00	10.33	625.21	6.42	500–1370

**Table 3 bioengineering-11-00837-t003:** Combined mean values and standard deviations (SDs) presented for all biochemical parameters tested on days 1 and 7, as there were no notable variations. Statistical evaluations were performed utilizing one- and two-way ANOVA.

Analyte	Blank Mean	Blank SD	C1 Mean	C1 SD	C2 Mean	C2 SD	References
ALB g/dL	4.56	0.04	4.92	0.10	5.02	0.11	4.1–5.4
TP g/dL	7.53	0.04	7.53	0.03	7.42	0.04	6.4–8.5
TB mg/dL	0.08	0.03	0.11	0.02	0.12	0.02	0.0–0.6
ALT U/L	31.71	0.46	32.86	1.10	31.49	0.75	26–37
ALP U/L	106.91	2.16	103.11	1.38	101.44	1.73	70–132
CREA mg/dL	1.07	0.04	1.22	0.07	1.15	0.06	0.5–1.4
UREA mg/dL	38.54	0.14	39.28	0.39	38.87	0.19	34.28–40.70
GLU mg/dL	133.12	1.04	130.27	1.00	129.14	0.42	114–143
CA mg/dL	11.47	0.12	11.76	0.14	11.61	0.04	10.5–13
PHOS mg/dL	9.43	0.13	9.13	0.40	9.10	0.54	5–13
K mmol/L	6.69	0.08	6.87	0.11	6.67	0.03	5.3–7.5
NA mmol/L	142.47	0.48	137.81	10.75	129.35	10.88	143–150

## Data Availability

The data are contained within the article.

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
