# Peer review of "The Potential of Composite Cements for Wound Healing in Rats"

_bioengineering, 2024, doi:10.3390/bioengineering11080837_

Round 1

Reviewer 1 Report

Comments and Suggestions for Authors

This paper, presenting the application of a composite characterized in the lab for skin regeneration by in vitro and in vivo experiments, is well written, designed and the results are well described. 

However, there are issues to solve:

1) there is no characterization of the biomaterial itself. Chemical, thermal and mechanical characterizations are missing. You mentioned also that you already tested the cytotoxicity, but you did not mention about biomaterial characteristics. 

2) Line 300-303, no images are present indicating what the authors reported. Please, provide them or delete it. 

3) No CFU counting or statistical data have been provided. why? 

4) Is it possible to merge Table 2 and 3 to indicate that there are no changes between Day1 and Day7?

5) same for Table 4 and 5 about biochemical profile. 

6) detection of macrophages and multinuclear macrophages indicating inflammation is a good and useful information. would it be possible provide a immunochemistry assay to stain, count them and compare the two composites? (I would suggest CD14, CD163 or TLR4)

7) Graph 2 and 3: units are missing. is it % right? please provide it. 

8) Graph 2 is about Day1 and Day 15. No MT images about Day1 are present to understand the angiogenesis occurring in C1 and the differences with C2 and Blank. Can you provide them please?

9) The authors discussed about the application of these materials used in dentistry. Did the authors ever think about using these materials in bone regeneration after osteolysis, osteoporosis or bone defects?

Comments on the Quality of English Language

English Language used is clear, but I suggest checking one more time before the submission to avoid typos and mistakes. 

Reviewer 2 Report

Comments and Suggestions for Authors

In the presented manuscript interesting research of employing nanohydroxyapatite in skin wound healing is presented. It is an in vivo study on rats, and histological and hematological testing were carried out. Even though interesting research, there are several unresolved issues.

I would suggest slight change of Title, more in spirit of English language.

Line 73 In an earlier study, we evaluated in vitro this two new composite cements. Please add the overall description of the explored cements, since you are mentioning them here for the first time in text.

Lines 78-79, 95-97 unclear sentences, with no clear intention and aim. English needs to be improved in whole manuscript.

The aim of the study is missed in Introduction, and sentences that more likely belong to the MM are set at the end of Introduction.

Line 128 swabs were obtained on the first and seventh day of the research. Swabs were taken from where, inside of the wounds, from the edges of the wounds? Which tissue, treatments?

Sections 2.2. and 2.3. should be after section 2.4.

2.4 Animal Care And Use - it should be the first section of MM

Line 226 Since the procedures didn't hurt, isoflurane ( Anesteran 99,9%, Rompharm, Otopeni, Romania) was employed - this sentence is not clear, you gave anesteran even though procedures were not painful?

Samples of tissue from the skin, liver, and kidneys were obtained at 7 and 15 days - how many animals were in the groups, considering 2-time frames, 5 animals for 7-day interval, and 5 animals for 15-day interval, in each group? Please clarify it for the readers.

Line 272 The mean ± SD (standard deviation) of the measured parameters were determined independently for days one and seven? Or 7 and 15? It is not clear.

Results are written with some elements of discussion in itself. Please to remove citations from Results and sentences that are more suitable for Discussion.

Line 300 The number of microorganisms reduced on the seventh day of the experiment, which was the only notable variation. - There is no numerical representation of this claim.

What was control group for hematological analysis? Previously in MM it is sad that ’’The biomaterials were then applied to the dermal defect as part of the protocol's next step. The left defect served as the control and no product was added while the experimental biomaterial was applied to the right defect. To facilitate the application, the composite cements C1 and C2 were synthesized as a pasta for this study’’. From this it could be concluded that all animals had one or other type of cement implanted in skin.

Cements were synthesized as paste for this study - cements were in the form of paste, and were further polymerized, after set in the wounds?

Graph 1. Legend is more common to be on the side of graph, usually to the right side. Why did you use one-way ANOVA and two-way ANOVA for statistics? Why not just two-way ANOVA? Question also applies for all other statistical testing. Also, did you check normality of data previous to statistical testing?

P for statistical significance is written as ’’p’’ and as ’P’’.

Overall, manuscript needs to be tight up and with consistent segments and descriptions. Also, English editing is needed so the manuscript reach higher quality.

Comments on the Quality of English Language

English editing is needed through whole text.

Reviewer 3 Report

Comments and Suggestions for Authors

Dear authors,

Thank you for giving me opportunity to review your manuscript entitled ' Composit cement research on wound healing in rats`.This study focused on the both  an organic and an in organic  matrix and evaluated the composite cement's ability to promote wound healing of the rat skin. The results of the study can present the possibility that the composit cement can be applied to regeneration therapy in the several tissues containing the skin. I think that the study has sufficient potential to be accepted in the journal. Please consider the following points to improve the contents in the manuscript.

Comment: The study mainly examined the ability of both hybrid cement composite C1 and C2 to improve the skin wound healing process. The authors demonstrated the results of the experiment using rat skin, kidney and liver. These results have been presented to clear.  In addition,  authors mentioned the ability of the hybrid composite cements for the bone regeneration therapy in the 'Abstract' and 'Discussion' . This study had not examined the biological effects of the hybrid composite cements, or the results on this matter have not been shown at all. I think that this content may be overspeculate.

Best regards

Round 2

Reviewer 1 Report

Comments and Suggestions for Authors

The authors answered to all the questions and for this reason, I consider the paper accepted and ready to be published. 

Reviewer 2 Report

Comments and Suggestions for Authors

Dear authors I would suggest different title, that would be more appropriate: Enhancing Rat Wound Healing with Composite Cements Application.

In the text there is still several typing errors, and English language needs to be improved prior to publishing.

Reference list should be unified.

Comments on the Quality of English Language

In the text there is still several typing errors, and English language needs to be improved.